# Research on the relationship between the scattering contribution and physical factors of the reference radiation regulated by ISO 4037–1

**Yi-kun Qian**[1], **Yi-xin Liu**[2], **Ben-jiang Mao**[2], **Song Zhang**[1], **Yanan Liu**[3], **Peng Feng**[1]*

**1** Key Laboratory of Optoelectronic Technology & Systems, Chongqing University, Chongqing, China, **2** Institute of Nuclear Physics and Chemistry, China Academy of Engineering Physics, Mianyang, Sichuan, China, **3** Department of Electronic Information Engineering, Chongqing Technology and Business Institute, Chongqing, China

* coe-fp@cqu.edu.cn

**Data Availability Statement:** All relevant data are within the manuscript and its Supporting Information files.

## Abstract

In the latest version of ISO 4037–1:2019 standard, the minimum dimension of a gamma radiation reference field was not clearly specified, which makes the construction of a mini-type gamma reference radiation field lack of scientific basis. This paper carried out the research on the relationship between the scattering contribution and physical factors of the reference radiation regulated by ISO 4037–1. LS-SVM was applied to construct the relational model between physical factors and scattering contribution based on the data simulated by Monte Carlo method. Then the minimum dimension of collimated reference radiation field is obtained by PSO algorithm. For Co-60 source, the minimum size of the radiation field obtained is 93 cm(L)×40 cm(W)×40 cm(H). For Cs-137 source, the minimum size of the radiation field obtained is 153 cm(L)×47 cm (W)×47 cm(H). The results meet the requirements of the standard based on the model and provides a technical reference for the design of a minitype reference radiation field.

## Introduction

To ensure the accuracy and reliability of gamma radiation dose (rate) meters, periodic calibration must be carried out in a standard gamma reference radiation [1]. The standard gamma reference radiation should be constructed in accordance with the requirements of the ISO 4037 [2–5] series standards. However, in the latest version of ISO 4037–1:2019, the description of the dimension of reference radiation is incomplete. For example, as illustrated in the standard "The distance from the export of the irradiation facility to the detector should be greater than or equal to 30 cm. The distance between the detector and the wall of the room along the beam direction should be sufficiently large for the contribution to the total air-kerma rate of photons back scattered by the walls of the room to be compatible with the requirements does not exceed 5%." The description "sufficiently large" was unclear, which brings inconvenience to build a minitype gamma reference radiation for movable application.

**Funding:** This work was partially supported by Chongqing Technological Innovation and Application Development Project(cstc2021jscx-gksbX0056), Chongqing Postgraduate Research and Innovation Project (CYB21059), Chongqing Basic Research and Frontier Exploration Project (cstc2020jcyj-msxmX0553)." The funders had no role in study design, data collection and analysis, decision to publish, or preparation of the manuscript.

**Competing interests:** The authors have declared that no competing interests exist.

Generally, in order to eliminate the influence of scattered photons in the radiation field, the dimension of an actual gamma radiation reference was meter scale. When the dimension of the radiation field is further reduced, there is no quantitative method to explore the relationship between the physical factors of the radiation field and the scattering contribution. At the same time, the application of movable reference radiation fields for the in-situ calibration work has become an important development trend. Hopewell Designs has developed the BX series calibration devices, providing general calibration functions of various portable or personal dosimeters. The dimension of the device is 181 cm(L) × 97 cm(W) × 206 cm(H). Liu [6–11] employed the machine prediction method based on sample instruments setting up a Mini-type Reference Radiation (MRR, a cube lead shielding box with length dimension of 60 cm), and successfully realized the determination of the conventional true value of air-kerma at the point of test. The measurement standard uncertainty was less than 4.6% and a movable calibration device has been developed successfully. The above work is mainly focused on the construction of minitype reference radiation fields. If the minimum dimension of the radiation field that meets the requirements of ISO 4037 series standards could be clearly specified, the development and use of movable reference radiation fields can be simplified and standardized.

In the previous research [12], the scattering contribution of each physical factor in a radiation reference field was clarified. However, the previous research has not studied the contribution of the combination of all physical factors. If different levels of all physical factors are combined for a comprehensive experiment, the amount of data to be simulated is huge, which is obviously undesirable.

Therefore, this paper applied the least squares support vector machines (LS-SVM) algorithm [13] to train a model between physical factors of reference radiation field and the scattering contribution based on the data simulated by Monte Carlo method. LS-SVM is machine learning method based on small sample data. From the above description, the various combination of all physical factors can be over ten thousand. Obviously, it is hard to simulate all data to build the model. Therefore, LS-SVM was a proper method to train a model with good robustness based on parts of sample data we simulated. Based on the model, the optimized design with minimum dimensions of the movable reference radiation field was obtained through the best combination of the physical factors. This involves the problem of comprehensive optimization of different parameters in their respective intervals. Particle swarm optimization (PSO) is a common parameter optimization method. The optimal solution can be found in the parameter space through the method, which is suitable for the application of this research. Therefore, it was applied to optimize the combination of all physical factors in the research.

## Methodology

### Simulation results of collimated reference radiation field

In this study, physical model of the collimated reference radiation field was built to achieve simulation data (Fig 1). Each experimental device in the field is placed on the central axis of the ray beam, and the radioactive source is placed in the geometric center of the container. The geometric center of the sensitive volume of the detector was placed coincide with the point of test in the reference radiation field.

Along the positive X direction of the spatial coordinate system, the distance between the radioactive source and the point of test is $D$. The distance between the final aperture and the detector is $D_1$. the distance between the point of test and the wall is $D_2$. Along the negative X direction of the spatial coordinate system, the distance between the radioactive source and the back wall is $D_3$. Along the Y direction of the space coordinate system, the distance between

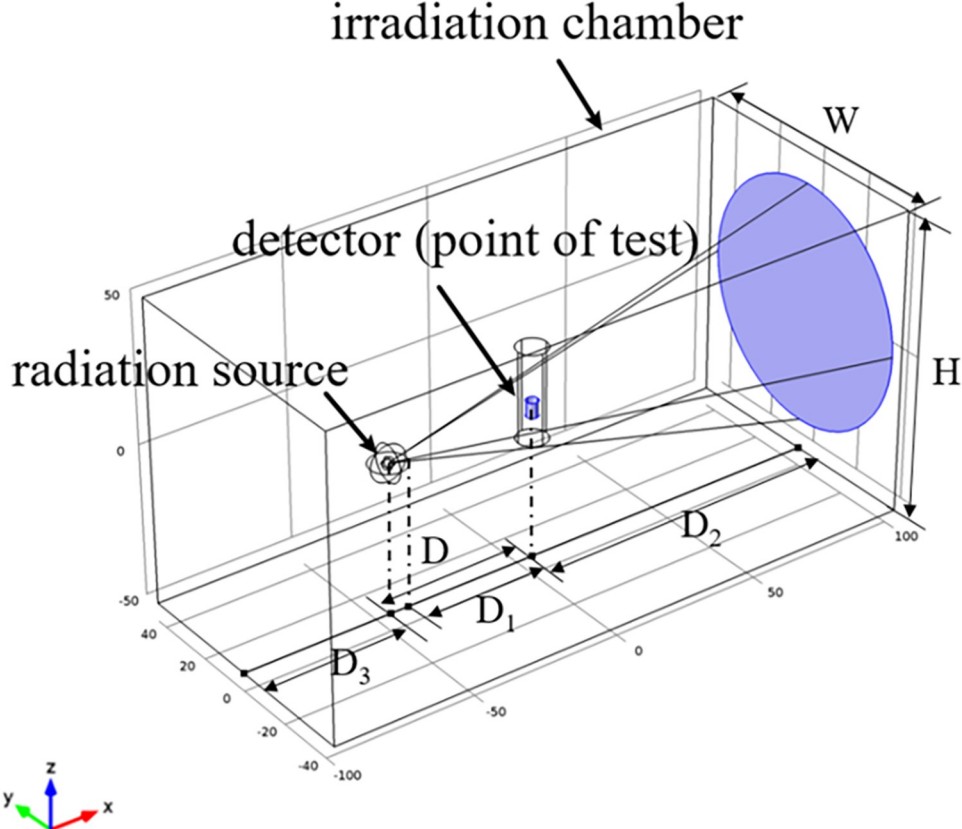

**Fig 1. Physical model of collimated reference radiation field.**

two walls of the irradiation room was defined as the width $W$. Along the Z direction of the space coordinate system, the distance between two walls of the irradiation room is defined as the height $H$.

In the previous simulation work [12], according to the size of the container specified by the standard and the minimum size of $D_1$, the value of $D$ was set as 66 cm. Furthermore, for the four physical factors of $D_3$, $D_2$, $W$, and $H$ in the model, the single factor rotation method is adopted. In each experiment, only one factor was changed while the others remain unchanged to calculate the energy spectrum at the point of test. Finally, influence law of scattering contribution caused by different physical factors is obtained.

The results showed that the wall surface along the ray beam directly will cause the scattering peak at the point of test, and the scattering contribution caused by the change of $D_2$ is the largest. For Co-60 and Cs-137, when the minimum dimensions of the collimated reference radiation field are 88 cm(L)×248 cm(W)×192 cm(H) and 104 cm(L) × 278 cm(W)×164 cm(H) respectively, the contribution of scattered photons reached 5% mentioned in the standard.

In fact, single factor rotation method is only suitable for the situation that no interaction existed between each two factors. However, when the size of the radiation field reduces sharply, the interaction between various physical factors of the radiation field was unknown. If different levels of all physical factors are combined through the exhaustive method, the total number of simulation experiment is extremely large. To obtain the correlation between the various physical factors of the radiation field within a limited number of experiments, and find the scattering contribution results at the point of test under the combined action of various

physical factors quickly, this paper uses the LS-SVM algorithm to construct the relational model between physical factors and the scattering contribution.

## Construction of the relational model between physical factors and scattering contribution

Support vector machine (SVM) is a machine learning method based on the Vapnik-Chervonenkis Dimension theory and the principle of structural risk minimization [14]. It seeks the optimal balance between the degree of model fit and the generalization ability, guarantees the global optimality of the quantitative solution results, and can better solve practical problems such as small sample classification and regression. However, the solution of quadratic programming is complicated, the method is usually combined with the least square method. The inequality constraints are transformed into equality constraints, and the complex nonlinear solution process is transformed into a linear matrix solution under the premise of ensuring accuracy, which expands the usability of the model [15].

With the simulation results in our previous work [12], a relational model based on LS-SVM was established as shown in Formula (1). $D_3$, $D_2$, $H$, $W$ are selected as the input vector. The physical factors $D$ and $D_1$ were set as 66 cm and 30 cm respectively. The scattering contribution from the source installation and the collimator system are included in the simulation data. The conventional true value of air-kerma at the point of test as the output vector.

$$y = f(D_3, D_2, H, W) \tag{1}$$

According to different physical factors that cause different degrees of scattering at the point of test [12], $D_3$, $D_2$, $H$, and $W$ take values in different ranges (Table 1).

The method of changing single variable was applied to determine $D_3$, $D_2$, $H$, and $W$ values (Table 1), $9^4$ sets of simulation experiment are needed. The workload of simulation is huge and difficult to achieve. In order to select the feature vectors in the modeling effectively and reduce the number of simulation experiment, this paper adopts the orthogonal experiment method [16]. According to the orthogonal table $L_{81}$ ($9^4$) of 4 factors and 9 levels of each factor, 81 groups of input vectors are obtained. Then use Monte Carlo simulation to get the conventional quantity value of air-kerma at the point of test when changing the values of $D_3$, $D_2$, $H$, and $W$. For the 81 groups of sample data obtained, the input is the size combination $x_i$ ($i = 1,2,3. ... ...,81$) of physical factors $D_3$, $D_2$, $H$, and $W$, and the output is the conventional quantity value $y_i$ ($i = 1,2,3. ... ...,81$) of air-kerma at the point of test. Each set of $x_i$ and $y_i$ is used as a set of sample data.

**Table 1. The values of input vector in the case of different radioactive sources.**

| Co-60 | | | | Cs-137 | | | |
|---|---|---|---|---|---|---|---|
| $D_3$/cm | $D_2$/cm | $H$/cm | $W$/cm | $D_3$/cm | $D_2$/cm | $H$/cm | $W$/cm |
| 13 | 1 | 40 | 40 | 7 | 5 | 40 | 40 |
| 14 | 3 | 50 | 50 | 8 | 8 | 50 | 50 |
| 15 | 5 | 60 | 60 | 9 | 12 | 60 | 60 |
| 16 | 7 | 70 | 70 | 10 | 16 | 70 | 70 |
| 25 | 20 | 80 | 80 | 25 | 20 | 80 | 80 |
| 50 | 30 | 100 | 100 | 50 | 30 | 100 | 100 |
| 100 | 50 | 150 | 150 | 100 | 50 | 150 | 150 |
| 200 | 100 | 200 | 200 | 200 | 100 | 200 | 200 |
| 400 | 300 | 300 | 300 | 400 | 300 | 300 | 300 |

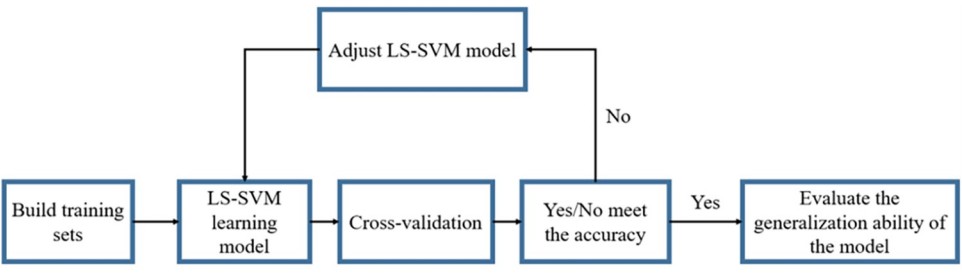

**Fig 2. Schematic diagram of model construction based on LS-SVM.**

The construction process of the LS-SVM relational model is mainly divided into the following parts (Fig 2):

A. Combining the sample data to obtain sample data matrices $x_{81\times4}$ and $y_{81\times1}$, and then distribute them to the training set and the validation set with a ratio greater than 1:1;

B. The kernel function of LS-SVM selects Gaussian radial basis function, initialize the regularization parameter $\gamma$ and kernel function width parameter $\sigma^2$ [17,18] of the training model, and train the relationship model $y = f(x)$ between $x_i$ and $y_i$;

C. Cross-validate the trained calculation model;

D. Evaluating the accuracy of the model;

E. If the accuracy does not meet the requirements, adjust the parameter $\gamma$ and $\sigma^2$ of the LS-SVM model, and retrain the relationship model $y = f(x)$ between $x_i$ and $y_i$;

F. If the accuracy meets the requirements, use the new test set to test the established model and evaluate the generalization ability of the model.

Due to the low complexity and less parameters of the trained model, the grid search method [19] is used to adjust the parameters $\gamma$ and $\sigma2$ of LS-SVM model in step E. The ranges of parameters $\gamma$ and $\sigma2$ was initialized and the possibilities of the two parameters were searched through loop traversal method.

In order to evaluate the accuracy of the relational model, the relative error and sample bias [20] are used to calculate the model solution error, as shown in Formulas (2) and (3).

$$e = \frac{|y_i - y_i'|}{y_i} \times 100\% \tag{2}$$

$$s = \sqrt{\sum_{i=1}^{N} (y_i - y_i')^2 / (N-1)} \tag{3}$$

Where $e$ is the relative error; $s$ is the sample bias; $N$ is the number of samples; $y_i$ is the dose rate at the point of test in the test set; $y_i'$ is the dose rate at the point of test achieved by the model.

## Optimization method for the minimum dimension of gamma reference radiation

To obtain a gamma reference radiation for movable application, the reference radiation should be as small as possible. Therefore, a radiation field with the smallest volume and the

contribution of scattered photons to the total air-kerma rate does not exceed 5% was applied as the optimal solution. How to optimize the parameters of the physical factors through an uncertain relationship between each two physical factors is the key technology to obtaining the optimal solution.

PSO is a parameter optimization method which iteratively produces optimal solutions through the cooperative of group solutions [21–23]. The solution of each optimization problem is a bird in the search space which called a particle. All particles have a fitness value determined by the optimized function. Speed and position are the only attributes of particles, indicating the movement speed and direction. The optimal solution of each individual searches in the search space separately, is recorded as the current individual extreme value. Compare the individual extreme value with other particles in the whole particle to find the optimal individual extreme value as the current global optimal solution for the whole particle swarm. All particles adjust their speed and position according to the current global optimal solution. Finally, the global optimal solution that satisfies the condition is output after a finite number of iterations.

The essence of using PSO to iteratively optimize the size of the radiation field is to optimize the input vector iteratively, which composed of the physical factors $D_2$, $D_3$, $H$ and $W$ of the radiation field. According to the size requirements of container, collimator and detector in the collimated reference radiation field in the previous simulation study [12], the values of the physical factors in the radiation field are set as: when the radioactive source is Co-60, the value range of $D_3$ is [13,30], the value range of $D_2$ is [5,400], the value range of $W$ is [30,400], and the value range of $H$ is [40,400]; When the radioactive source is Cs-137, the value range of $D_3$ is [7,30], the value range of $D_2$ is [5,400], the value range of $W$ is [30,400], and the value of $H$ The range is [40,400]. The population number $m$ is set to 50, the iteration number $n$ is 500, the inertia weight $w$ is 1, and the learning factors $c_1$ and $c_2$ are 2.

In the optimization process, the contribution of scattered photons to the total air-kerma rate does not exceed 5% is used as a constraint condition. The fitness is defined as the volume of the minitype reference radiation field. It is obvious that the volume of the reference radiation field is smaller, the fitness is lower.

The optimization design of minitype reference radiation by PSO algorithm mainly includes the following steps (Fig 3):

A. Set the parameters in the PSO algorithm, and initialize the particle group;

B. The trained LS-SVM model was applied to calculate the conventional true value P of air-kerma at the point of test of each individual;

C. Judged whether the output P obtained by each individual brought into the LS-SVM model meets the constraint condition of P<5%. If it was satisfied, calculated the fitness of the individual; if not, assigned the fitness of the individual to $10^8$ through the penalty function;

D. Run the PSO algorithm to optimize and update the speed and position of each individual;

E. When the iteration number reached the requirement, the optimal solution was outputted.

## Results and discussion

### Relational model trained based on LS-SVM

In order to test the accuracy and generalization capabilities of the trained LS-SVM relational model, the Monte Carlo method was used to simulate the collimated reference radiation field of 50 scenes of different sizes. Various size factors of the simulated scene are all taken value

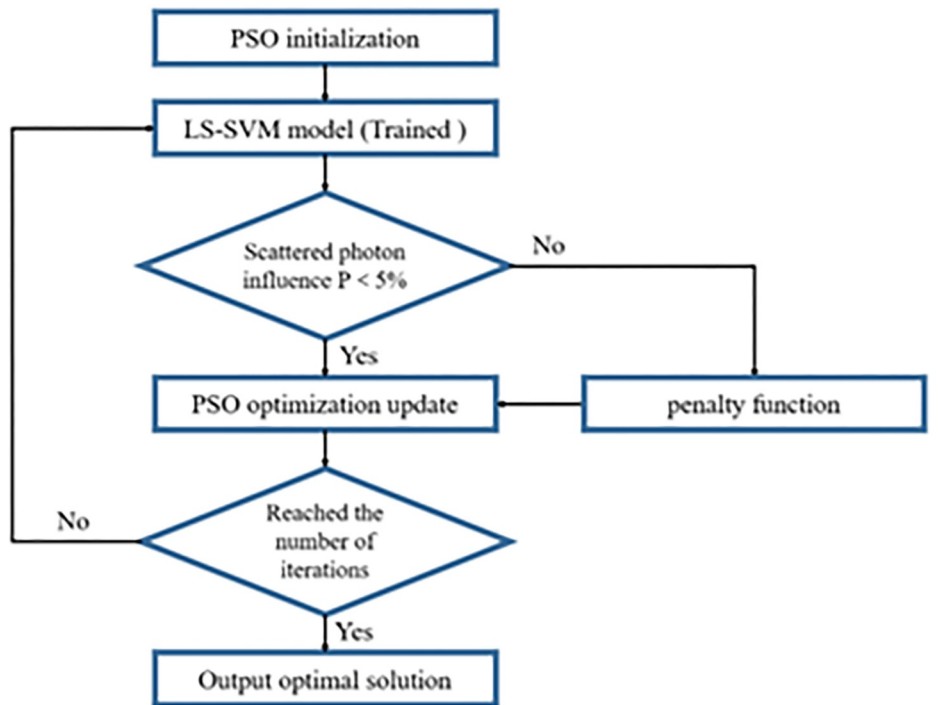

**Fig 3. Optimized design process of minitype reference radiation.**

within the value range (Table 1), which is no need to take the specific value in Table 1. The simulation results are compared with the model results (Fig 4).

By comparing the simulation results and model solutions, the accuracy of the relationship model is evaluated. When the radioactive source was Co-60, the average error and sample deviation of the solution results were 0.40% and 0.075 respectively. The average correction deviation of the scattering contribution caused by the prediction error is less than 0.16%. When the radioactive source was Cs-137, the average error and sample deviation of the solution results were 1.04% and 0.028 respectively. The average correction deviation of the scattering contribution caused by the prediction error is less than 0.21%. The results showed that the

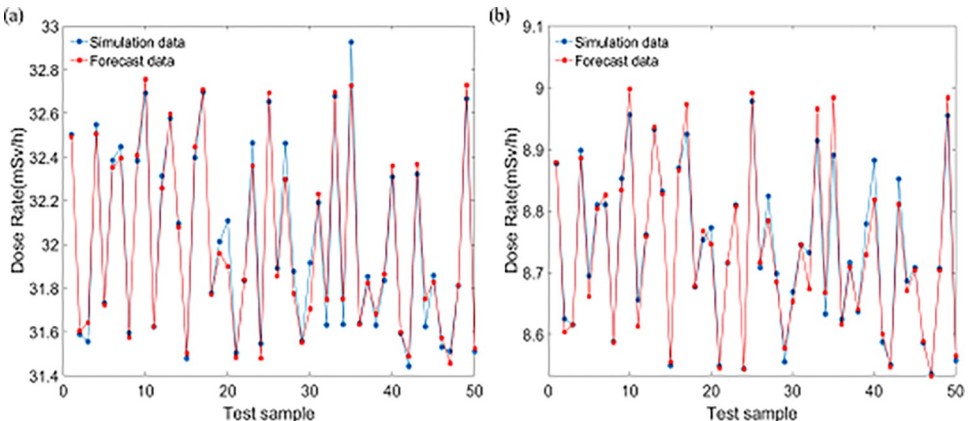

**Fig 4.** Comparison between simulation and the model results: (a) Co-60, (b) Cs-137.

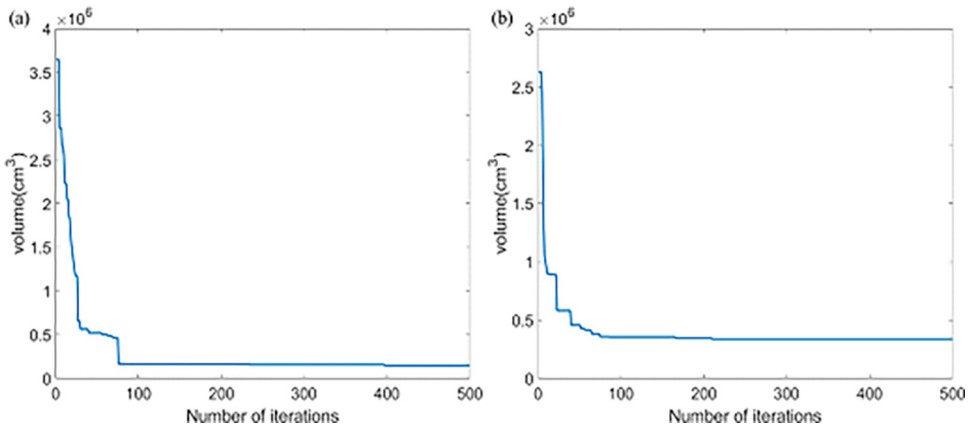

**Fig 5.** The fitness mean values of the global best particles: (a) Co-60, (b) Cs-137.

LS-SVM relational model has high calculation accuracy and can be used to establish and improve the gamma reference radiation field.

## Optimized combination of physical factors through PSO method based on the trained model

After getting the trained relational model, the combination of physical factors can be optimized by PSO algorithm. With the increase of iteration times, the fitness gradually tends to be stable (Fig 5).

The results show that as the number of iterations gradually increases, the fitness drastically drops. When the radioactive source was Co-60, the minimum volume of the radiation field is 0.1488 m$^3$. The global optimal solution corresponding $D_3$, $D_2$, $W$ and $H$ were 13 cm, 14 cm, 40 cm and 40 cm respectively. When the radioactive source was Cs-137, the minimum volume of the radiation field was 0.3380 m$^3$. The global optimal solution corresponding $D_3$, $D_2$, $W$ and $H$ were 7 cm, 80 cm, 47 cm and 47 cm respectively. In order to test the above results, the radiation fields of the same size were simulated. The scattering contribution at the point of test are 4.98% and 4.99% respectively, which meets the requirement that the contribution of scattered photons to the total air-kerma rate does not exceed 5%. Compared with the previous results [12], $D_2$ increases slightly but $W$ and $H$ are greatly reduced. This further confirms that the effects of different physical factors are not independent.

## Conclusions

In view of the fact that it is not yet clarified in the ISO 4037–1:2019 to what extent the radiation field size is reduced to still meet the scattering requirements, this paper applied LS-SVM to obtain a relationship model for the radiation field size and the influence from scattered rays in the radiation field: when the radioactive source is Co-60, the mean error and sample deviation of the solution results are 0.40% and 0.075 respectively. The mean correction deviation of the scattering contribution caused by the solution error is less than 0.68%. When the radioactive source was Cs-137, the mean error and sample deviation of the solution results are 1.04% and 0.028 respectively. The mean correction deviation of the scattering contribution caused by the solution error was less than 1.13%.

Based on the trained LS-SVM, the PSO algorithm was used to iteratively optimize the size of each physical factor, and the minimum size of the collimated reference radiation field that

meets the standard requirements is obtained. When the radioactive source was Co-60, the minimum size of the radiation field obtained is 93 cm(L)×40 cm(W)×40 cm(H). When the radioactive source was Cs-137, the minimum size of the radiation field obtained was 153 cm (L)×47 cm (W)×47 cm(H). Compared with the previous results [12], the volume of the radiation field is reduced by 96.45% and 92.87% respectively.

This paper combined the LS-SVM relational model with the PSO algorithm and proposed an optimization design method for minitype reference radiation field. This work effectively overcome the shortcomings of the single-factor rotation method when there are interactions between various factors, and has high calculation accuracy and strong generalization. Since the LS-SVM model was not complicated, the method was used in practice with transferability (when encountering similar geometries, in particular, different sizes of $D_1$ and containers). It provides a technical reference for the design of reference radiation field, especially the minitype reference radiation field.

## Supporting information

**S1 Dataset.**
(ZIP)

**S1 File.**
(ZIP)

## Acknowledgments

This work is a collaboration between Chongqing University College of Optoelectronic Engineering and Institute of Nuclear Physics and Chemistry. The authors would like to thank Xin Liu for excellent technical support and Professor Biao Wei for critically reviewing the manuscript.

## Author Contributions

**Funding acquisition:** Peng Feng.

**Investigation:** Song Zhang.

**Methodology:** Ben-jiang Mao.

**Validation:** Yi-xin Liu.

**Visualization:** Yi-xin Liu.

**Writing – original draft:** Yi-kun Qian.

**Writing – review & editing:** Yanan Liu, Peng Feng.

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
