## [Decision Letter · Decision Letter 0]

2 Jun 2022

PONE-D-22-12755Research on the relationship between the scattering contribution and physical factor of the reference radiation regulated by ISO 4037-1PLOS ONE

Dear Dr. Feng,

Thank you for submitting your manuscript to PLOS ONE. After careful consideration, we feel that it has merit but does not fully meet PLOS ONE’s publication criteria as it currently stands. Therefore, we invite you to submit a revised version of the manuscript that addresses the points raised during the review process.

We look forward to receiving your revised manuscript.

Kind regards,

Ashwani Kumar, Ph.D.

Academic Editor

PLOS ONE

Journal Requirements:

"This work was partially supported by Chongqing Technolog

ical Innovation and Application Development Project(cstc2021jscx-gksbX0056),

Chongqing Postgraduate Research and Innovation Project (CYB21059), Chongqi

ng Basic Research and Frontier Exploration Project (cstc2020jcyj-msxmX0553)."

Reviewers' comments:

Reviewer's Responses to Questions

**Comments to the Author**

1. Is the manuscript technically sound, and do the data support the conclusions?

Reviewer #1: Yes

Reviewer #2: Yes

2. Has the statistical analysis been performed appropriately and rigorously? 

Reviewer #1: Yes

Reviewer #2: Yes

3. Have the authors made all data underlying the findings in their manuscript fully available?

Reviewer #1: No

Reviewer #2: Yes

4. Is the manuscript presented in an intelligible fashion and written in standard English?

Reviewer #1: Yes

Reviewer #2: Yes

5. Review Comments to the Author

Reviewer #1: 1- Introduction: Is well organized; however, it should be a summary of the literature review. Additionally, there is no comprehensive description of why the method is used?

2- Equations (2) and (3) need references.

3- In Table 1 how did choose the value of the input vector?

4- In the construction process of the LS-SVM relational model, the stage of adjusting the LS-SVM model is not clear and needs to explain (Fig 2).

5- The paper should be rewritten as the following:1-introduction 2-methodology 3-results-4-discussion 5-Conclusion.

6- You need to correct many grammatical mistakes.

Reviewer #2: The topic and Objective set out in the work is clear. The methodology used is also appropriate and the final results have been clearly spelt out. The work clearly contribute the some missing knowledge as far as the use of ISO 4037-1:2019 is concerned

6. PLOS authors have the option to publish the peer review history of their article (what does this mean?). If published, this will include your full peer review and any attached files.

Reviewer #1: No

Reviewer #2: **Yes: **Raymond Agalga

---

## [Author Response · Author response to Decision Letter 0]

29 Aug 2022

Dear Reviewers,

Thank you very much for your time involved in reviewing the manuscript and your very encouraging comments on the merits.

We also appreciate your clear and detailed feedback and hope that the explanation has fully addressed all of your concerns. In the remainder of this letter, we discuss each of your comments individually along with our corresponding responses.

To facilitate this discussion, we first retype your comments in italic font and then present our responses to the comments. 

We would like to take this opportunity to thank you for all your time involved and this great opportunity for us to improve the manuscript. We hope you will find this revised version satisfactory.

Yours sincerely,

Peng Feng

 

Responses List to Reviewer 1

Comment 1:

Introduction: Is well organized; however, it should be a summary of the literature review. Additionally, there is no comprehensive description of why the method is used?

Response 1: 

Thanks for your comment. In the revision, we revised the content of the introduction part. In the last two paragraphs, we added the reason why LSSVM method was used for modeling and PSO algorithm was used for optimization.

Comment 2:

Equations (2) and (3) need references.

Response 2: 

In the revision, reference 'Mathematical statistics with applications' for equations (2) and (3) was added.

Comment 3:

In Table 1 how did choose the value of the input vector? 

Response 3: 

Thanks for your comment. Table 1 shows the value of the input vector under different radioactive sources. If the univariate control method is used to select the values of D3, D2, H, and W in Table 1, 94 groups of simulation experiments are required. The workload of simulation is huge and difficult to achieve. In order to effectively select the eigenvectors in the modeling and reduce the number of simulation experiments, the orthogonal experiment method is adopted here: According to the orthogonal table L81 (94) of 4 factors and 9 levels of each factor, 81 sets of input vector groups are obtained.

In the revision, we supplement the method of selecting the input vector，in the paragraph below Table 1.

Comment 4:

In the construction process of the LS-SVM relational model, the stage of adjusting the LS-SVM model is not clear and needs to explain (Fig 2).

Response 4: 

Thanks for the comment. In the revision, we have revised and supplemented the modeling process of LSSVM, in the paragraph below Figure 2. 

In the stage of adjusting LS-SVM model, the parameters γ and σ2 need to be adjusted. Because the model needs to be established is low in complexity and few parameters, the grid search method is used here: setting the value range of parameters γ and σ2, trying every possibility through loop traversal, and finally getting the best combination of parameters. 

Comment 5:

The paper should be rewritten as the following:1-introduction 2-methodology 3-results-4-discussion 5-Conclusion.

Response 5: 

Thank you for the comment. In the revision, we adjusted the structure of the paper into four parts: 1-Introduction; 2-Methods; 3-Results and discussion; 4-Conclusion.

Comment 6:

You need to correct many grammatical mistakes.

Response 6: 

Thank you for the detailed review. We have carefully and thoroughly proofread the manuscript to correct all the grammar and typos. 

Responses List to Reviewer 2

Comment 1:

The topic and Objective set out in the work is clear. The methodology used is also appropriate and the final results have been clearly spelt out. The work clearly contribute some missing knowledge as far as the use of ISO 4037-1:2019 is concerned

Response 1:

Thank you for your positive comments.

---

## [Decision Letter · Decision Letter 1]

2 Dec 2022

Research on the relationship between the scattering contribution and physical factors of the reference radiation regulated by ISO 4037-1

PONE-D-22-12755R1

Dear Dr. Feng,

We’re pleased to inform you that your manuscript has been judged scientifically suitable for publication and will be formally accepted for publication once it meets all outstanding technical requirements.

Kind regards,

Ashwani Kumar, Ph.D.

Academic Editor

PLOS ONE

Additional Editor Comments (optional):

Reviewers' comments:

Reviewer's Responses to Questions

**Comments to the Author**

1. If the authors have adequately addressed your comments raised in a previous round of review and you feel that this manuscript is now acceptable for publication, you may indicate that here to bypass the “Comments to the Author” section, enter your conflict of interest statement in the “Confidential to Editor” section, and submit your "Accept" recommendation.

Reviewer #1: All comments have been addressed

2. Is the manuscript technically sound, and do the data support the conclusions?

Reviewer #1: Yes

3. Has the statistical analysis been performed appropriately and rigorously? 

Reviewer #1: Yes

4. Have the authors made all data underlying the findings in their manuscript fully available?

Reviewer #1: Yes

5. Is the manuscript presented in an intelligible fashion and written in standard English?

Reviewer #1: Yes

6. Review Comments to the Author

Reviewer #1: The authors have adequately addressed my comments raised in a previous round of review and I feel that this manuscript is now acceptable for publication

7. PLOS authors have the option to publish the peer review history of their article (what does this mean?). If published, this will include your full peer review and any attached files.

Reviewer #1: No

---

## [Editor Report · Acceptance letter]

19 Dec 2022

PONE-D-22-12755R1 

Research on the relationship between the scattering contribution and physical factors of the reference radiation regulated by ISO 4037-1 

Dear Dr. Feng:

I'm pleased to inform you that your manuscript has been deemed suitable for publication in PLOS ONE. Congratulations! Your manuscript is now with our production department. 

Kind regards, 

on behalf of

Dr. Ashwani Kumar 

Academic Editor

PLOS ONE